# Novel variable neighborhood search heuristics for truck management in distribution warehouses problem

Akram Y. Sarhan[1], Loai Kayed B. Melhim[2], Mahdi Jemmali[3,4,5,6], Faycel El Ayeb[7,8], Hadeel Alharbi[9] and Ameen Banjar[10]

[1] Department of Information Technology, College of Computing and Information Technology at Khulis, University of Jeddah, Jeddah, Saudi Arabia
[2] Department of Health Information Management and Technology, College of Applied Medical Sciences, University of Hafr Al Batin, Hafr Al Batin, Saudi Arabia
[3] MARS Laboratory, University of Sousse, Sousse, Tunisia
[4] College of Computing and Informatics, University of Sharjah, Sharjah, United Arab Emirates
[5] Department of Computer Science and Information, College of Science at Zulfi, Majmaah University, Al-Majmaah, Saudi Arabia
[6] Department of Computer Science, Higher Institute of Computer Science and Mathematics, Monastir Uuniversity, Monastir, Tunisia
[7] Unit of Scientific Research, Applied College, Qassim University, Saudi Arabia
[8] GRIFT Research Group, CRISTAL Laboratory, National School of Computer Sciences, La Manouba University, Manouba, Tunisia
[9] Department of Information and Computer Science, College of Computer Science and Engineering, University of Ha'il, Hail, Saudi Arabia
[10] Department of Information Systems and Technology, College of Computer Science and Engineering, University of Jeddah, Jeddah, Saudi Arabia

Corresponding author
Akram Y. Sarhan, asarhan@uj.edu.sa

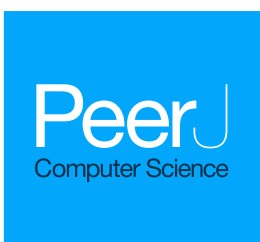

## ABSTRACT

Logistics and sourcing management are core in any supply chain operation and are among the critical challenges facing any economy. The specialists classify transport operations and warehouse management as two of the biggest and costliest challenges in logistics and supply chain operations. Therefore, an effective warehouse management system is a legend to the success of timely delivery of products and the reduction of operational costs. The proposed scheme aims to discuss truck unloading operations problems. It focuses on cases where the number of warehouses is limited, and the number of trucks and the truck unloading time need to be manageable or unknown. The contribution of this article is to present a solution that: (i) enhances the efficiency of the supply chain process by reducing the overall time for the truck unloading problem; (ii) presents an intelligent metaheuristic warehouse management solution that uses dispatching rules, randomization, permutation, and iteration methods; (iii) proposes four heuristics to deal with the proposed problem; and (iv) measures the performance of the proposed solution using two uniform distribution classes with 480 trucks' unloading times instances. Our result shows that the best algorithm is $\widetilde{OIS}$, as it has a percentage of 78.7% of the used cases, an average gap of 0.001, and an average running time of 0.0053 s.

## INTRODUCTION

Logistics and supply chains rely on commodities being stored in distribution facilities. Distribution centers are critical for successfully and swiftly delivering items within supply chains (*Syabilla & Mulyaningtyas, 2023*). As a result, these industries must have efficient systems in place to manage large numbers of trucks and unload their shipments efficiently (*Engesser et al., 2023*; *Müller, 2023*).

The amount of global commercial exchange has expanded in recent years. According to the World Trade Organization report on April 5, 2023, global trade growth in 2022 was 2.7% and is predicted to grow by 1.7% in 2023 (*World Trade Organization, 2023*). This expansion will result in increasing demand for goods, overcrowding, fierce rivalry between businesses, and product delivery delays. It also causes many issues, including unpredictable demand, sustainability, ethical issues, higher transportation costs, and raw material shortages, which might jeopardize supply chains.

Furthermore, due to the frequent changes in competitive marketplaces, customer satisfaction has become critical for organizations in the logistics and cargo distribution systems domains. Customers demand on-time delivery of products and services (*Cui et al., 2023*). Many firms are using current and established supply chain management theories to change logistics and commodities distribution systems due to the significance of reacting within the boundaries of the available time to ensure user satisfaction with the delivery time (*Li & Li, 2023*). As a result, numerous industries recognized the significance of supply networks. Disruption in the supply chain causes disarray in all aspects of modern life. As a result, companies worldwide are looking deeper at their global supply chains and the technology that powers them, wondering what they can do to future-proof their businesses.

Supply chain operations require different transport companies to handle thousands of transport operations continuously to ensure the continuity of supply operations and the non-interruption of supply chains for industrial, commercial, medical, and other domains. In these operations, various goods are transported from the main points to the distribution points across broad geographical regions. For the success of these operations, transportation operations are entrusted to an enormous number of trucks, which are constantly moving to transport these goods to the storage areas.

Different transported goods lead to various loads and loading times. Therefore, the greater the demand for goods, the greater the transported quantity, and the more trucks will transport these goods. As the number of trucks increases, the difficulty of distributing those trucks to unload them will increase, which increases trucks' unloading time and causes delays in supply operations. This delay is undesirable and will lead to more challenges to the sustainability of supply operations. Companies realized this effect and employed all available technologies to develop and improve supply chains (*Syabilla & Mulyaningtyas, 2023*). Improved supply chains and logistics management transform businesses, where companies become more competitive by minimizing costs and increasing efficiency. Correct management of supply chains also ensures the timely delivery of products (*Li & Li, 2023*).

Businesses utilize trucks to guarantee the sustainability of their supply chains by transporting various goods and crucial resources required for uninterrupted production and satisfied customers. Trucks play a crucial role in ensuring that warehouses are well-stocked with products ready to be distributed to customers.

The successful operation of supply chains necessitates efficient coordination among various transportation modes: air, sea, and land. The primary benefits of road transportation stem from the exclusive access trucks have to an extensive infrastructure that other modes of transportation cannot match. The road network stands out as the most important transport infrastructure by a large margin. Furthermore, the transportation of goods by road is not reliant on hubs such as ports, airports, or train stations, and there are few destinations that cannot be accessed *via* road. Trucks provide advantages for sea, air, and rail transportation as well. In the majority of instances, there is a need for supplementary road transportation to transport goods between the airport, sea ports, train stations, and plants or warehouses, or *vice versa*. The diversity and adaptability of transporting goods by road provide boundless opportunities for moving them between different locations.

Despite the employment of modern technologies, the increase in demand requires more transportation operations, supply operations, logistical support, and infrastructure, which requires more costs. Increased costs reduce competitiveness and weaken the growth opportunities of various businesses (*Chen, Golhar & Banerjee, 2023*; *Zahid, Khurshid & Ying, 2023*). These factors explain the complexity of the problem and increase the need for innovative solutions that will maximize the use of available resources to reduce the high cost.

This research presents the truck unloading problem with a predetermined unloading time and a limited number of warehouses to distribute all incoming trucks for unloading in the available warehouses in the shortest possible time. Interest in this type of problem began after the great demand of consumers to buy online, especially during COVID-19. Since that time, distributors needed storage much more than marketing or direct selling. Increasing inventory requires moving more merchandise to warehouses, which accumulates more waiting trucks in warehouse yards. The truck waiting event added extra time for haulage operations, negatively impacting supply operations.

The suggested solution will employ variable neighborhood search to solve the presented problem, which has impressive performance and a low execution time. The proposed method is an enhancement tool that can boost the performance of any given heuristic. It uses perturbations to derive a better solution. The proposed approach will solve approximately the presented NP-hard problem. The effect of the proposed system on the speed of securing the strategic stockpile is not tangible in normal circumstances. However, this effect increases tremendously during exceptional events (*Dobler, 2013*) such as severe weather conditions, wars, epidemics, and other similar conditions.

The limitations of the proposed solution can be summarized as follows: the system provides an approximate solution to the presented problem. Finally, the system was not tested with large-scale instances.

The rest of this article is organized as follows: an analysis of previous and similar works is presented in "Literature Review". The description of the presented problem is exposed in "Problem Description". While the details of the proposed solution, the variable neighborhood search algorithm, and the used initial solutions are presented in "The Variable Neighborhood Search (VNS) Metaheuristic". The discussion of the obtained results is viewed in "Results and Discussion". Finally, "Conclusion" presents the conclusion and future work.

## LITERATURE REVIEW

Transportation using trucks as one of the major carriers in logistics operations have attracted the attention of many researchers because of the impact of this sector on supply chain operations, so the problem of scheduling trucks within a set of determinants and criteria is one of the problems that have been raised repeatedly among researchers. But this type of problem involves many challenges and difficulties. To clarify the difficulty of dealing with the problem of truck scheduling in supply chain and logistics management processes, especially the problems in which certain trucks are assigned to a specific door, the researchers in *Fabry et al. (2022)* studied the complexity of this type of problem, and the results they reached showed that such problems are NP-hard in general, even when dealing with only one door.

Therefore, many researchers have tried to employ various methods and models to solve this problem; for example, *Heidari, Zegordi & Tavakkoli-Moghaddam (2018)* used two meta-heuristics to solve the problem of scheduling incoming and outgoing trucks in a multi-berth facility with unknown vehicle arrival times. The proposed model aimed to deal with the uncertainty and non-deterministic nature of truck arrival times. *El Hachemi, Gendreau & Rousseau (2013)* utilized two phases of heuristics to solve truck scheduling problems based on given constraints in the forest industry. The first phase determines the destinations of full truckloads. While the second phase routes and schedules the daily transportation of logs using two methods.

*Nossack & Pesch (2013)* used two-stage heuristic solution approach to solve the truck scheduling problem in intermodal container transportation between customers and various terminals with strict time window constraints enforced by the stack holders. The objective was to minimize the total truck operating time while considering the enforced constraints. The cross-dock truck scheduling problem was presented by *Xi et al. (2020)* as a two-stage optimization model, which was solved by a column and constraint generation algorithm. Extensive numerical experiments validated the efficiency of the developed algorithm.

Mixed-integer programming model with an Adaptive Particle Swarm Optimization algorithm was used by *Hop, Van Hop & Anh (2021)* to solve the quay crane and yard truck scheduling problem, the authors aimed to minimize the total time required to finish the unloading and transport operations of all assigned containers. From a different perspective, *Skaf et al. (2021)* presented the scheduling problem of a single-dock crane and multi-yard trucks to reduce the total completion time of transporting all containers from the container ship to its storage location. The researchers solved the scheduling problem by

developing a genetic algorithm and obtained a near-perfect solution with reasonable CPU time.

The hybrid genetic algorithm with variable neighborhood search was presented by *Fan et al. (2019)* to solve the truck scheduling problem when there is an outdoor yard and several main gates to reduce the number of trucks used, reduce the total cost, and improve task scheduling and delivery efficiency. The research problem presented by *Fan et al. (2019)* is similar in some aspects to the research problem presented in this context, with differences in the solution method, assumed constraints, and the main objective. In an attempt to speed up the unloading of trucks in the truck scheduling problem, *Tadumadze et al. (2019)* made an integrated scheduling of trucks and manpower to speed up the unloading of trucks, where the researchers used manpower management and simultaneous planning by providing models with mixed integers that integrate manpower planning and truck scheduling. The researchers show that integrated planning can significantly increase truck scheduling performance in terms of total time and punctuality.

Solving such problems requires overcoming critical challenges other than time and reliability, such as profit. For this reason, many industries seek the help of technology to reduce the overall costs of supply chain operations. *Melchiori et al. (2022)* used a mixed integer linear programming model to optimize transportation issues in the forest industry. The proposed model handles transportation characteristics such as allocation, routing, and scheduling. In a different approach, *Fathollahi-Fard et al. (2019)* modified the Social Engineering Optimizer to find an optimal condition to solve the truck scheduling problem for receiving and shipping truck sequences for large-scale cases. The Taguchi experimental design technique improved the optimization performance of the created algorithms.

In the review presented by *Theophilus et al. (2019)*, the researchers presented a state-of-the-art review of articles that are concerned with the problem of truck scheduling at distribution centers, with special attention to the main features of distribution operations. A set of challenges were extracted, and some ideas for future research were proposed.

In supply chain management and e-procurement processes, *Jemmali, Melhim & Alharbi (2019)* presented a customized algorithm based on the decision maker's or customer's preferences to solve the problem of suitable supplier selection. Various solution methods, strategies, and models show the extent of attention given to logistics operations and supply chain management (*Romagnoli et al., 2023*; *Liu et al., 2023*; *Naim & Gosling, 2023*). Benefiting from technical developments to build solutions that will serve people and improve their lives is one of the highest goals of scientific research. Therefore, researchers in this work have developed many intelligent solutions to handle various sectors' problems, such as in public health (*Jemmali, 2022*; *Jemmali et al., 2022c*; *Melhim, 2022*), developing smart cities and enhancing provided services (*Jemmali et al., 2022b*; *Jemmali, 2021*).

To complete the efforts exerted in various fields and the participation of researchers for their belief in the importance of humans, the authors developed a set of algorithms that can maximize the minimum completion time to finish all assigned tasks and optimize the given systems (*Jemmali et al., 2022a*; *Jemmali, Melhim & Al Fayez, 2022*; *Melhim, Jemmali & Alharbi, 2018*). Such systems may be adapted to assist the concerned authorities in identifying the scope of the problem and calculating the magnitude of the consequent

damage as soon as possible. Therefore, in this research, the truck scheduling problem, in a parking lot and some warehouses was raised as a contribution from researchers to develop existing solutions and build new solution structures that contribute to the development of supply chains and achieve more prosperity and stability for humanity.

Despite the effectiveness of neighborhood search algorithms in handling scheduling problems, only some researchers have dealt with these tools compared to other algorithms or solving models. Many researchers employed variants of neighborhood search algorithms to address significant problems in various domains. For example, *Serna et al. (2021)* presented a global-local neighborhood search algorithm to model the flexible job shop scheduling problem. The experimental evaluation of the proposed model showed satisfactory performance.

A variable neighborhood search heuristic was employed in different domains with different models. For example, *Ayedi (2023)* applied an improved meta-heuristic algorithm for multi-relay underground wireless sensor networks to optimize the network resource efficiency and lifetime extension. *Zhang & Chen (2022)* used it to minimize total lagging and quality risk, where the branch-and-bound algorithm was used as a search operator to handle parallel-machine scheduling problems. Although the approach presented by *Zhang & Chen (2022)* employed a variable neighborhood search heuristic, it is not similar to the approach presented in this context. The proposed algorithms can be adapted to be applied on the systems studied in *Melhim, Jemmali & Alharbi (2018)* and *Jemmali, Alharbi & Melhim (2018)*.

To the best of our knowledge, the problem at hand has never been studied in the literature. The truck management systems identified in the literature addressed the problem of truck scheduling based on route distribution, traveled distances, and delivery times, while the suggested approach is concerned with scheduling waiting trucks in the parking yard in front of warehouses to be unloaded at the assigned warehouses. The proposed solution employs variable neighborhood search to solve the presented problem. Variable neighborhood search has impressive performance and a short execution time. Furthermore, it is an enhancement tool that can boost the performance of any given heuristic.

# PROBLEM DESCRIPTION

To simplify the supply chain management problem, this section introduces a set of concepts that clarifies the different aspects of the problem. The sub-sections include problem presentation, modeling, research constraints, research value, and the mathematical formulation of the problem.

## Problem presentation

Due to the extent of the impact of trucks on supply chains and logistical operations, the problem of truck scheduling has found great interest, whether in the research and academic aspect by researchers and scientists or the economic and commercial aspect by industries and companies. Typically, the number of trucks waiting in the warehouse yard competing to unload their cargo in time is greater than the number of available warehouse

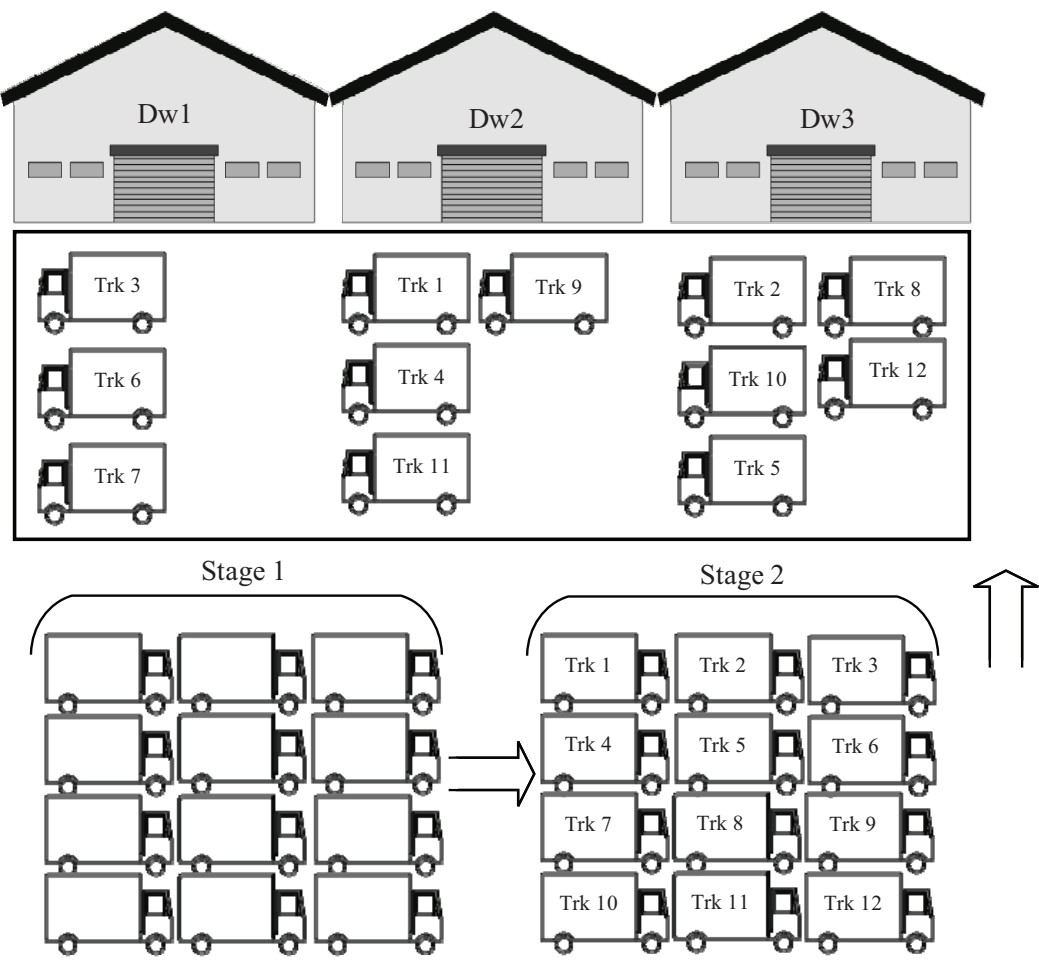

**Figure 1 General overview of the truck scheduling system.**

doors, forcing some trucks to wait longer for other trucks to finish, as shown in Fig. 1. This delay may have effects other than the delay in delivery time, such as affected supply chains, stopping some industries, economic losses, changing some cargo characteristics such as fresh foods, and other effects. Truck scheduling assigns each incoming truck to the warehouse door and sets a time to unload the truck so that the management system of these trucks is optimized.

Therefore, in the proposed model, incoming trucks are received in a big yard, as shown in Fig. 1, where a unique identifier is given to each truck. This identifier is used by the developed intelligent algorithms to determine the warehouse number that the truck should go to. The truck unloading mechanisms available in all warehouses are similar, so the chosen warehouse number will not affect the unloading time, and each warehouse receives only one truck at a time. The main goal that the algorithms should achieve, is to distribute all incoming trucks to the available warehouses in a way that ensures the unloading of all trucks in the least possible time.
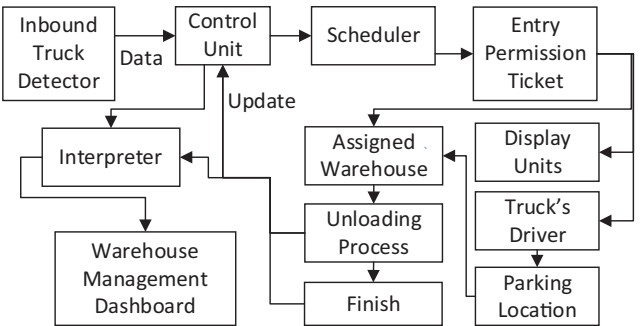

**Figure 2 Trucks unloading block diagram.**

## Modeling

The block diagram of the proposed system is shown in Fig. 2. This system is composed of a set of integrated parts that work together, which includes the following:

- Inbound Truck Detection: collect real-time data about the trucks that enter the warehouse yard, then assign a unique identifier to each truck based on truck entrance time and plate number.
- Control Unit: The control unit processes the received data to specify the number of existing trucks in the warehouse yard at a certain time. It performs the necessary processing to provide the scheduler with the updated information to derive the new schedules.
- Scheduler: This part of the system uses the information provided by the control unit to generate a schedule that uses the developed metaheuristic to assign vehicles to different warehouses. The generated schedule defines the destination of each truck by specifying the warehouse number to which each truck must drive.
- Entry Permission Ticket: Based on the scheduler output, this part creates an entry permission ticket containing the truck identifier, truck plate number, entry time, and truck parking lot number in front of the warehouse. The generated permission will be in the form of a printed ticket delivered to the truck driver and in an electronic file to be displayed on dedicated screens on the fronts of the warehouses.
- Display Units: Display units are located in selected locations to ensure maximum visibility to truck drivers. Through these units, the identifier of trucks assigned to a specific warehouse is displayed, in addition to directions and updates that ensure smooth distribution and movement of trucks to the locations that are supposed to be reached. Furthermore, these units also update drivers with various information, such as the remaining time till unloading time, which helps drivers to take advantage of this time.
- Vehicle Driver: The truck driver must comply with the instructions and information printed on his entry permit and displayed on the various display screens to ensure that the truck is in the right position for the unloading process to start.

- Parking Location: The place where all incoming trucks will be lined up, as the space in front of the warehouses is used to provide parking lots for trucks in a way that facilitates trucks' access to the assigned warehouses. The parking location is linked with the warehouse number specified for each truck, noting that this location will continue to change until the truck reaches its designated warehouse.

- Assigned Warehouse: The warehouse in which the truck unloading process will take place, where the warehouses are provided with a copy of the trucks' entry permission to enable the truck distribution management system to determine the unloading warehouse for each truck and the total number of unloaded trucks for each warehouse. When the truck arrives at the specified warehouse, the truck data is matched with the stored data in the warehouse management system then the unloading system is instructed to start the process of unloading trucks.

- Unloading Process: Once the truck arrives at the unloading spot in the specified warehouse, the process of truck unloading begins after taking instructions from the warehouse management system, then the data of the unloaded truck is recorded, and a copy of that data is sent to the control unit to update the scheduler's data and to the interpreter to get the information needed for the system dashboard.

- Interpreter: analyzes the incoming data to obtain all information related to truck unloading operations in all warehouses, such as the total number of inbound trucks, the total number of unloaded trucks, the status of all warehouses, the number of trucks allocated to each warehouse, the number of remaining trucks, the number of trucks that have left and the number of trucks which is being unloaded now, in addition to any other information related to the trucks or the warehouse management system in general. This information is sent to the system dashboard.

- Warehouse Management Dashboard: aims to control the unloading operations and monitor the warehouse management system to make quick interventions when necessary and issues the required reports on the warehouses' status and trucks unloading operations.

## Research constraints

The proposed solution uses four algorithms as an initial solution to the developed heuristic to solve the presented model. In this solution, each warehouse has a single gate, the truck unloading time is the same as the loading time, various loads have different loading times and thus the unloading time, and the style of unloading is assumed to be the same for all warehouses and the warehouse can receive only one truck at a time.

## Research value

The end customer may not feel the importance of supply chains and the size of the operations carried out to ensure the flow of goods and the continuity of their presence on the shelves. Any delay or obstruction at any stage of the various supply chain processes may significantly impact the consumer, as it is in all sectors. However, in exceptional circumstances such as epidemics, lockdowns, floods, extreme weather conditions,

earthquakes, and volcanoes, supply operations become very sensitive, critical and directly affect people's lives. Therefore, the proposed solution will significantly improve the performance of supply chains, whether in normal circumstances or even in exceptional circumstances, and the role of this system may be more important in exceptional circumstances due to the increased importance of timing in critical circumstances.

Minimizing the total needed time to unload all inbound trucks means the ability of the system to receive more trucks without any extra infrastructure for the unloading operations. Moreover, this reduces costs and increases profits. More inbound trucks mean more goods, which increases the stock without additional costs and enhances the supply chain process in general. Moreover, when the trucks finish unloading, they are free to go. Thus, they can perform more tasks instead of wasting more time while waiting for their turn for the unloading process to start.

## Mathematical formulation

The problem can be formulated as follows. Given a set of trucks $Trk$ to unload their shipments in the available distribution warehouses $Dw$. The total number of trucks is $n_{Tr}$, and the number of distribution warehouses is $n_{Dw}$. The operating time to unload a truck $j$ is denoted by $Ot_j$. The cumulative operating time when a truck $j$ is unloaded is denoted by $Co_j$. After finishing the unloading of all trucks, the total operating time in the distribution warehouse $i$ is denoted by $To_i$. The maximum total operating time of all distribution warehouses is denoted by $To_{max}$ and given in Eq. (1).

$$To_{max} = \max_{1 \leq i \leq n_{Dw}} To_i \tag{1}$$

The maximum total operating time $To_{max}$ can be expressed as given in Eq. (2).

$$To_{max} = \max_{1 \leq j \leq n_{Tr}} Co_j \tag{2}$$

The goal is to finish unloading all trucks with a minimum operating time. This goal is reached when the developed algorithms find a schedule that minimizes $To_{max}$. Minimizing $To_{max}$ is an NP-hard problem.

## THE VARIABLE NEIGHBORHOOD SEARCH (*VNS*) METAHEURISTIC

This section presents the variable neighborhood search (*VNS*) algorithm and the four algorithms that were used as initial solutions for the *VNS*. Indeed, in the first step, an initial solution is determined by applying one of the four proposed algorithms presented in the next subsection. After that, the call of *VNS* based on this initial solution will give a new solution. Finally, the best value is selected. This procedure is repeated $n_{Tr} - 1$ times. The details of the *VNS* procedure are given in the "The variable neighborhood search (*VNS*) algorithm".

---

**Algorithm 1** Second initial solution (*FIS*) algorithm.

1: Set $LG(Trk)$

2: Call $Sch(Trk)$

3: Calculate $To^1_{max}$

4: Set $SM(Trk)$

5: Call $Sch(Trk)$

6: Calculate $To^2_{max}$

7: Calculate $To_{max} = min(To^1_{max}, To^1_{max})$

8: Return $To_{max}$

---

## The four-used initial solutions

A set of four algorithms were used as initial solutions to solve the presented problem. These four algorithms are based on the dispatching rules and randomization methods. The based methods have proven their simplicity and efficacy in several scheduling problems. In addition, these methods have given a good result in a remarkable execution time.

The first algorithm is based on the dispatching rules approach, where we apply two sorts of methods to arrange $Ot_j$ and then schedule the defined trucks to the distribution warehouses. The best solution will be selected. The second algorithm is based on the selection of the longest $Ot_j$ value, after finishing the scheduling of $n_{Tr} - 1$ trucks, assign the selected truck. For the third algorithm use the same method as the second algorithm but use the longest $Ot_j$, after that select the smallest $Ot_j$. The last algorithm is based on the randomized method. Indeed, the probabilistic procedure is applied to select between the two longest $Ot_j$ values.

- First initial solution (*FIS*): This algorithm is based on the dispatching rule. The first step in this algorithm is to arrange the $Ot_j$ values according to the non-increasing order. Then, assign the truck of interest to the most available distribution warehouse, which is the warehouse that has the minimum $To_i$. After that, the obtained completion time is denoted by $To^1_{max}$. The second step is to arrange the $Ot_j$ values according to the non-decreasing order. Then, schedule the truck of interest to the most available distribution warehouse. The obtained completion time is denoted by $To^2_{max}$. The minimum value between $To^1_{max}$ and $To^2_{max}$ is returned. To summarize, $To_{max} = min(To^1_{max}, To^1_{max})$. The function that returns the list $L$ that is sorted according to the non-increasing order, is dented by $LG(L)$, while the function that returns the list $L$ that is sorted according to the non-decreasing order will be denoted by $SM(L)$. The function $Sch(L)$ is responsible for scheduling the truck values in $L$ to the most available distribution warehouses. The instructions of *FIS* are detailed in Algorithm 1.

- Second initial solution (*SIS*) This algorithm is based on the following. First, search for the truck that has the longest $Ot_j$. This truck is denoted by $Tr^L$, then apply *FIS* algorithm on the remaining $n_{Tr} - 1$ trucks. Finally, assign the truck $Tr^L$ to the most available distribution warehouse. The function that returns the longest element in the list $L$ is

---

| **Algorithm 2** Second initial solution (*SIS*) algorithm. |
|---|
| 1: Set $Tr^L = LGT(Trk)$ |
| 2: Set $TrkL = Trk \setminus Tr^L$ |
| 3: Call *FIS(TrkL)* |
| 4: Call $SchM(Tr^L)$ |
| 5: Calculate $To_{max}$ |
| 6: Return $To_{max}$ |

| **Algorithm 3** Third initial solution (*TIS*) algorithm. |
|---|
| 1: Set $Tr^S = LGS(Trk)$ |
| 2: Set $TrkS = Trk \setminus Tr^S$ |
| 3: Call *FIS(TrkS)* |
| 4: Call $SchM(Tr^S)$ |
| 5: Calculate $To_{max}$ |
| 6: Return $To_{max}$ |

denoted by $LGT(L)$. The function $SchM(X)$ is responsible for scheduling the truck $X$ to the most available distribution warehouse. The instructions of *SIS* are detailed in Algorithm 2.

- Third initial solution (*TIS*) This algorithm is based on the following. First, search for the truck that has the smallest $Ot_j$. This truck is denoted by $Tr^S$. After that, apply *FIS* algorithm to the remaining $n_{Tr} - 1$ trucks. Finally, assign the truck $Tr^S$ to the most available distribution warehouse. The function that returns the smallest element in the list $L$ is denoted by $LGS(L)$. The instructions of *TIS* are detailed in Algorithm 3.

- Fourth initial solution (*OIS*) This algorithm is built as follows. First, select the two trucks that have the longest $Ot_j$ values. Then, between these two trucks, select the one that is scheduled in the most available distribution warehouse. This selection is based on probabilistic choice. Indeed, we fix a probability $\theta$. The choice of the first truck with the longest $Ot_j$ is applied with a probability $\theta$. However, the second truck with the largest $Ot_j$ is applied with probability $1 - \theta$. Continue scheduling until finishing all trucks. This procedure is repeated 500 times. The function that returns an element in the list $L$ satisfying the probability $\theta$ is denoted by $STeta(L)$. The instructions of *OIS* are detailed in Algorithm 4.

## The variable neighborhood search (*VNS*) algorithm

The variable neighborhood search (*VNS*) algorithm shown in Algorithm 5, is based on the permutation of the given solution to find a new solution, which is better than the initial solution. Suppose that the algorithm denoted by *AlG* solves the studied problem with a

---

**Algorithm 4** Fourth initial solution *(OIS)* algorithm.

1: Set *LG(Trk)*

2: **for** ($j = 1$ to $n_{Tr}$) **do**

3:     Call $TrB = STeta(Trk)$

4:     Call *Sch(TrB)*

5:     Set $Trk = Trk \setminus TrB$

6: **end for**

7: Calculate $To_{max}$

8: Return $To_{max}$

---

**Algorithm 5** Variable neighborhood search *(VNS)* algorithm.

1: Call $Sq = SQA(Tr)$

2: Calculate $To^0_{max} = AlG(Tr)$

3: **for** ($k = 1$ to $n_{Tr} - 1$) **do**

4:     Call $SwP(Sq, k, k+1)$

5:     Calculate $To^k_{max} = AlG(Sq)$

6: **end for**

7: Calculate $To_{max} = \min_{0 \leq k \leq n_{Tr} - 1} To^k_{max}$

8: Return $To_{max}$

---

schedule that is denoted by $Sh_0$. Relatively, the procedure $AlG()$ is responsible to calculate and return the $To_{max}$ value after applying $AlG$. The first step of the proposed *VNS* is to determine the sequence of the obtained schedule denoted by $Sq_0$. Then, denote by $SQA()$ the function that returns the sequence of the schedule applying the algorithm $AlG$. The latter sequence will be permuted by the *VNS* function with a new schedule and sequence denoted by $Sh_1$ and $Sq_1$. A new solution to the studied problem is derived and returned based on the latter schedule and sequence. After that, take $Sq_1$ and apply a permutation on it to find another new schedule, and so on, until the number of permutations is less than $n_{Tr} - 1$. The function that returns a new list $V$ after swapping the element $a$ and $b$ in the list $V$ given as input is denoted by $SwP(V, a, b)$.

After applying the proposed *VNS* on the *FIS, SIS, TIS*, and *OIS* the result of each algorithm is denotes by $\widetilde{FIS}$, $\widetilde{SIS}$, $\widetilde{TIS}$, and $\widetilde{OIS}$.

## RESULTS AND DISCUSSION

In this research, the developed algorithms were implemented using C++. The used machine has a Windows 10 operating system with Core i5 6200 CPU @ 2.30 GHz 2.40 2GHz and a RAM of 8.00 GB. The use of the dynamic memory and pointer in the developed code for the proposed algorithms can affect the running time when the operating system is changed. This research will choose two classes to be tested. The

**Table 1 Number of trucks and warehouses' distribution.**

| $n_{Tr}$ | $n_{Dw}$ |
|---|---|
| 10, 20 | 2, 3, 5 |
| 30, 50, 100 | 2, 3, 5, 10 |
| 200, 250 | 10, 15, 20 |

**Table 2 Overview of all algorithms in *Prg* and *Time*.**

| | *FIS* | *SIS* | *TIS* | *OIS* | $\widetilde{FIS}$ | $\widetilde{SIS}$ | $\widetilde{TIS}$ | $\widetilde{OIS}$ |
|---|---|---|---|---|---|---|---|---|
| *Prg* | 18.1% | 0.0% | 18.5% | 42.1% | 31.3% | 0.2% | 34.7% | 78.7% |
| *AG* | 0.026 | 0.088 | 0.021 | 0.012 | 0.015 | 0.074 | 0.012 | 0.001 |
| *Time* | 0.0000 | 0.0000 | 0.0000 | 0.0010 | 0.0000 | 0.0000 | 0.0001 | 0.0053 |

selected classes have a different style in generating $Ot_j$. The uniform distribution $U(60, 150)$ is the first class to be chosen, while the uniform distribution $U(20, 100)$ is the second chosen class. The selected total number of trucks $n_{Tr}$ and the selected total number of distribution warehouses $n_{Dw}$ is shown in Table 1.

In this context, 10 instances of each class were generated for $n_{Tr}$ and $n_{Dw}$ values. Table 1 shows the number of the total generated instances $(2 \times 3 + 3 \times 4 + 2 \times 3) \times 10 \times 2 = 480$.

To measure the performance of the developed algorithms, the following indicators will be tested.

- $\hat{T}$ the best $T_c$ the value returned after all algorithms are executed.
- $T$ the $T_c$ the returned value when executing the studied heuristic.
- $G = \frac{T-\hat{T}}{\hat{T}}$.
- *AG* is the average of $G$ over a number of instances
- *Prg* the percentage of instances when $T = \hat{T}$ is reached.
- *Time* the average running time of the algorithm. The time is calculated in seconds. We are denoted by "." if the time is less than 0.001 s.

The overall results of the developed algorithms are shown in Table 3. It is obvious that the best algorithm is $\widetilde{OIS}$, as it has a percentage of 78.7% of the used cases, an average gap of 0.001, and an average running time of 0.0053 s. While the algorithm *SIS* is the worst with a percentage result of 0.0%, an average gap of 0.088, and an average running time which is less than 0.001 s. Table 2 shows the impact of the proposed neighborhood search.

The average gap $G$ performance values when changing the number of trucks $n_{Tr}$ is shown in Table 3. The given results showed that increasing the number of trucks $n_{Tr}$ does not affect the performance of the developed algorithms even with large $n_{Tr}$ values. For example, the average gap values of $\widetilde{OIS}$ algorithm reached a value of less than 0.001 when

**Table 3 The performance in *AG* and *Time* for all algorithms when the number of trucks changed.**

| $n_{Tr}$ | FIS | SIS | TIS | OIS | $\widetilde{FIS}$ | $\widetilde{SIS}$ | $\widetilde{TIS}$ | $\widetilde{OIS}$ |
|---|---|---|---|---|---|---|---|---|
| 7 | 0.003 | 0.083 | 0.003 | 0.000 | 0.001 | 0.078 | 0.001 | 0.000 |
| 8 | 0.035 | 0.122 | 0.019 | 0.008 | 0.014 | 0.095 | 0.006 | 0.000 |
| 9 | 0.014 | 0.070 | 0.014 | 0.001 | 0.003 | 0.056 | 0.004 | 0.000 |
| 10 | 0.046 | 0.111 | 0.028 | 0.019 | 0.025 | 0.091 | 0.015 | 0.001 |
| 15 | 0.034 | 0.093 | 0.026 | 0.012 | 0.019 | 0.071 | 0.014 | 0.000 |
| 20 | 0.034 | 0.076 | 0.030 | 0.018 | 0.020 | 0.064 | 0.011 | 0.001 |
| 25 | 0.031 | 0.087 | 0.027 | 0.019 | 0.021 | 0.075 | 0.019 | 0.001 |
| 30 | 0.037 | 0.090 | 0.027 | 0.026 | 0.028 | 0.079 | 0.021 | 0.001 |
| 40 | 0.020 | 0.077 | 0.020 | 0.013 | 0.013 | 0.069 | 0.013 | 0.002 |
| 50 | 0.010 | 0.053 | 0.010 | 0.005 | 0.004 | 0.046 | 0.004 | 0.003 |

**Table 4 The performance in *AG* and *Time* for all algorithms when the number of distribution warehouses changed.**

| $n_{Dw}$ | FIS | SIS | TIS | OIS | $\widetilde{FIS}$ | $\widetilde{SIS}$ | $\widetilde{TIS}$ | $\widetilde{OIS}$ |
|---|---|---|---|---|---|---|---|---|
| 2 | 0.027 | 0.085 | 0.027 | 0.014 | 0.016 | 0.059 | 0.013 | 0.000 |
| 3 | 0.027 | 0.078 | 0.014 | 0.004 | 0.008 | 0.065 | 0.005 | 0.001 |
| 4 | 0.023 | 0.116 | 0.020 | 0.015 | 0.019 | 0.107 | 0.013 | 0.001 |
| 5 | 0.044 | 0.078 | 0.021 | 0.012 | 0.015 | 0.068 | 0.014 | 0.002 |
| 8 | 0.002 | 0.085 | 0.031 | 0.026 | 0.031 | 0.070 | 0.016 | 0.002 |

$n_{Tr} = \{7, 8, 9, 15\}$, while the only other algorithm that reached a value of less than 0.001 is OIS when $n_{Tr} = 7$.

The average gap $G$ performance values when changing the number of distribution warehouses $n_{Dw}$ is shown in Table 4. The given results showed that increasing the number of distribution warehouses $n_{Dw}$ has some effect on the performance measurements of the developed algorithms. For example, the average gap values of the $\widetilde{OIS}$ algorithm reached a value of less than 0.001 when $n_{Dw} = 2$.

Figure 3 illustrates the average gap variation for $\widetilde{FIS}$ and $\widetilde{SIS}$ algorithms when the pair $(n_{Tr}, n_{Dw})$ changes. The given figure shows a similar variation change in the relation for the two algorithms with a notice that the average gap variation of $\widetilde{FIS}$ is smaller than that of $\widetilde{SIS}$ algorithm.

Figure 4 illustrates the average time variation for $\widetilde{OIS}$ algorithms the pair $(n_{Tr}, n_{Dw})$ changes. Logically, as the number of trucks and warehouses increases the required tasks greatly increase, this is what can be noticed from Fig. 4. When the value of pair $(n_{Tr}, n_{Dw})$ goes beyond 24 a gradual increase in time required to complete all tasks can be observed. This increase explains that increasing the number of warehouses is not always the perfect solution for such problems.

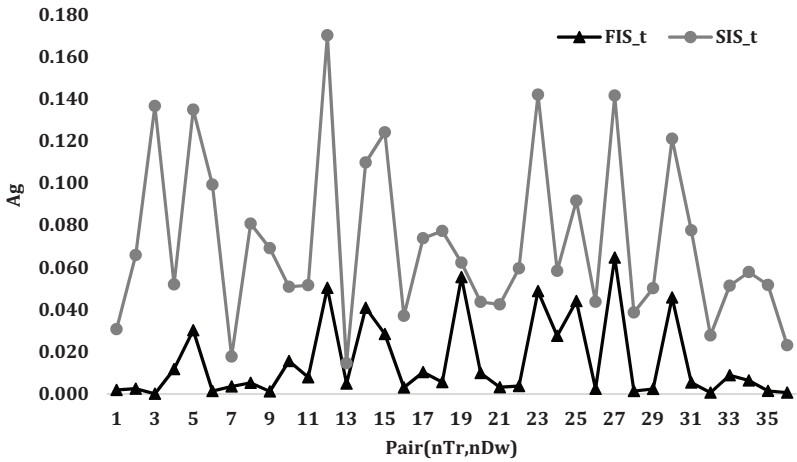

**Figure 3 The average gap variation for $\widetilde{FIS}$ and $\widetilde{SIS}$ algorithms the pair $(n_{Tr}, n_{Dw})$ changes.**

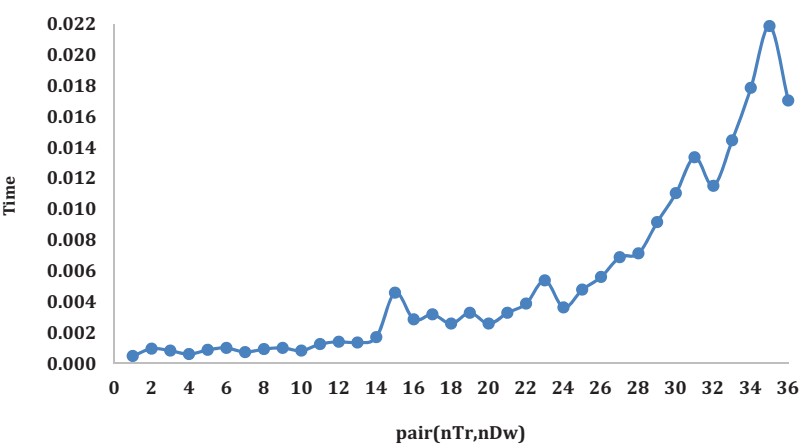

**Figure 4 The average time variation for $\widetilde{OIS}$ algorithms the pair $(n_{Tr}, n_{Dw})$ changes.**

## CONCLUSION

The operations of supply chains and logistical services maintain the continuous flow and arrival of goods to the user efficiently and effectively at a competitive price. Logistics usually uses fleets of various transportation means, especially trucks. Trucks are the primary mode of transportation for many areas of logistical services and supply chains. However, this industry faces many challenges that will affect the smoothness of supply chains and system efficiency, which causes prices to increase. Traditional solutions contribute to resolving these challenges, but with more infrastructure and expenditures, which have a direct effect on the operational costs of managing the supply systems and the goods' basic prices, which will be a wrong choice in light of the intense competition between companies to control prices and obtain consumer satisfaction. This research presented the challenge of unloading waiting trucks by allocating them to different warehouses within a set of determinants, the most significant of which is to solve the

problem by leveraging existing infrastructure to prevent expensive costs while maintaining the highest level of provided services. This problem is solved by a creative warehouse management system that uses an intelligent metaheuristic with four algorithms as an initial solution to achieve the desired goals. Two uniform distribution classes generated 480 instances of various trucks' unloading times. These instances evaluated the performance of the proposed system. Experimental results obtained by the tested indicators show that the best algorithm was $\widetilde{OIS}$, as it has a percentage of 78.7% of the used cases, an average gap of 0.001, and an average running time of 0.0053 s.

The future work of this research may revolve around three directives. First, the presented model should be enhanced by changing the initial solutions to derive new results. Subsequently, the suggested model should be implemented on comparable issues in diverse real-world sectors, like healthcare (*Melhim, 2022*), forest monitoring (*Jemmali et al., 2023*), and railways (*Jemmali, Melhim & Al Fayez, 2022*). Finally, a lower bound solution of the proposed model should be developed and subsequently how far the proposed solutions are from the derived lower bound should be computed.

### Funding
This work was funded by the deputyship for Research & Innovation, Ministry of Education in Saudi Arabia through the project number (MoE-IF-UJ-22-4100335-2). The funders had no role in study design, data collection and analysis, decision to publish, or preparation of the manuscript.

### Grant Disclosures
The following grant information was disclosed by the authors:
Research & Innovation, Ministry of Education in Saudi Arabia: MoE-IF-UJ-22-4100335-2.

### Competing Interests
The authors declare that they have no competing interests.

### Author Contributions
- Akram Y. Sarhan analyzed the data, authored or reviewed drafts of the article, and approved the final draft.
- Loai Kayed B. Melhim performed the experiments, analyzed the data, performed the computation work, authored or reviewed drafts of the article, and approved the final draft.
- Mahdi Jemmali conceived and designed the experiments, performed the experiments, analyzed the data, performed the computation work, prepared figures and/or tables, and approved the final draft.
- Faycel El Ayeb performed the experiments, prepared figures and/or tables, and approved the final draft.
- Hadeel Alharbi conceived and designed the experiments, performed the computation work, prepared figures and/or tables, and approved the final draft.

- Ameen Banjar analyzed the data, performed the computation work, prepared figures and/or tables, authored or reviewed drafts of the article, and approved the final draft.

## Data Availability

The raw data and code are available in the Supplemental Files.

## Supplemental Information

Supplemental information for this article can be found online at http://dx.doi.org/10.7717/peerj-cs.1582#supplemental-information.

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
