# Peer review of "Novel variable neighborhood search heuristics for truck management in distribution warehouses problem"

_PeerJ Computer Science, doi:10.7717/peerj-cs.1582_

## Round 0.1 · original submission · Major Revisions

Dear authors,

Your paper has been reviewed by two reviewers who asked for revisions of the paper. Please revise the paper according to comments by reviewers, mark all changes in new version of the paper and provide cover letter with replies to them point to point.

Reviewer 1 ·

Basic reporting

1. English writing should be improved. There are a lot of long and confusing sentences (such as "This paper presents supply chain optimization, by discussing the problem of scheduling trucks’ unloading operations, for the case of a limited number of warehouses, one yard, and one door for each warehouse, given that the number of trucks and the truck unloading time is different and known in advance."). Divide sentences like this into two or three separate sentences. Other than that, spelling, grammar, syntax, and style errors need to be addressed.
2. Literature review does not cover all aspects of the problem and methodology. For example, there is no review of the studies dealing with variable neighborhood search, which is one of the main aspects of the paper. In addition, the authors did not identify any research gaps that their study is trying to cover.
3. The paper is too partitioned. Is it necessary to have sub-sub-sections 4.1.1, 4.1.2, etc? They consist of only a few sentences.
4. Figure 2 is not mentioned anywhere in the main text. All figures must be quoted somewhere in the main text.
5. In line 310 the authors quote Table 2 instead of Table 1.

Experimental design

6. It is unclear from the abstract what are the main contributions of this paper. This should be clearly implied.
7. Introduction is not written well. The authors should clearly establish the main motives, aim, and research questions. in addition the authors should highlight the the paper's main results, conclusions, and contributions. In addition, at the end of the introduction, the authors should provide a brief overview of the following sections (this is not mandatory but is considered a standard in academic writing).
8. The author did not provide a mathematical formulation of the problem. They only offered notation. The authors must provide enough information to enable the replication of the results.
9. Why four heuristics? Why not more or less? This is not justified well enough.
10. Heuristics are not explained well enough.

Validity of the findings

11. The authors should highlight what is novel in their methodology compared to the previous studies.
12. The paper does not have a proper discussion. The authors did not discuss how the results can be interpreted from the perspective of previous studies. Discussion should clearly and concisely explain the significance of the obtained results to demonstrate the article's actual contribution to this field of research when compared with the existing and studied literature. In addition, the discussion should point out the theoretical and practical implications of the study.
13. The authors did not propose any future research directions. They should provide at least 3-5 solid future research directions that would interest most of the Journal readership.

Additional comments

14. There are no keywords.
15. There are certain technical issues:
a) There should be at least a couple of sentences between headings of different levels (e.g. between section 3 and sub-section 3.1).
b) Some of the references are missing volume (and issue) numbers.
c) References are not quoted in the main text appropriately. They are just placed at the end of the sentence. Place them in the brackets (e.g. instead of "... their shipments efficiently Engesser et al. (2023)Muller (2023)." write "... their shipments efficiently (Engesser et al., 2023; Muller, 2023).")

Reviewer 2 ·

Basic reporting

1. The call to references is not correct and creates confusion when reading.
2. Please improve the writing of the manuscript. I recommend paying more attention to the clarity of expression and readability, namely sentence structure, length, and others.
3. The authors claim: „The volume of trade exchange in the world has increased dramatically over the past few years“ (lines 49-51).
It would be important to state how many times or to what degree
4. The authors claim: „...delays in providing products also lead to the emergence of several problems that may threaten supply chains“ (lines 51, 52).
Which ones? Please name at least some of key
5. The authors argue that with the increase in the number of trucks, the difficulties in the distribution system increase by increasing the time required for unloading, thus creating delays. They then state the effect (lines 71-76).
What kind of effect is it about? In addition, there are studies in the literature that are not exactly consistent with this statement.

Experimental design

1. The description of the problem is clear, and the great importance of the functioning of the supply chain is indicated. However, it should be clearly specified what effects are achieved by minimizing the waiting time of trucks for unloading.
2. What information is processed by the Control Unit?
3. Does the proposed model respect the limitations related to the type of goods? Are there priorities in unloading trucks?
4. Given the article deals with the allocation of trucks in the warehouse with the aim of minimizing the time for unloading, it should be explained more clearly in the part about the research value.

Validity of the findings

1. In the first algorithm, the two trucks with the longest Ot values are selected, and the one scheduled at the distribution warehouse with the highest available is then given preference. Please describe the parameters by which the most available distribution warehouse is identified.
2. What impacts and significance does the performance of the equipment and operating system have on the reliability of the results?
3. The authors need to discuss the limitations of the proposed method as well as the limitations of the case study. What are your recommendations for future investigations? How has the proposed method been implemented in the case study problem?
4. In the Conclusion Section, I did not find limitations, research gaps, or future research recommendations.
5. How can practitioners use the proposed method to solve real-life problems, and how is the proposed method useful for future studies?
6. Is the "Proposed Model" a novel or new approach? In its absence, would the results be significantly different?

Additional comments

This study is interesting and provides certain answers to the permanently present challenges in supply chains. However, it needs notable improvements, which are commented on through suggestions and questions.

---

## Round 0.2 · Minor Revisions

Dear author,

Your revised paper has been reviewed by two reviewers and one of them asked for revisions of the paper. Please revise the paper according to comments by reviewer, mark all changes in new version of the paper and provide cover letter with replies to them point to point.

Reviewer 1 ·

Basic reporting

All issues from the previous review round have been resolved.

Experimental design

All issues from the previous review round have been resolved.

Validity of the findings

All issues from the previous review round have been resolved.

Additional comments

All issues from the previous review round have been resolved.

Reviewer 2 ·

Basic reporting

Dear authors,
significant progress has been noted in your research. Contemporary literature relevant to the research question was used. Numerical indicators also confirmed the importance of the analyzed problem. Despite significant progress compared to V0, this research requires certain improvements, especially in the domain of the role of distribution centers in supply chains. In the literature used, there are acceptable explanations of this importance, and they should be adequately interpreted in the paper. It would therefore be useful to enrich your research through suggestions and answers to questions:
- "Distribution stations" is a phrase used in relation to the electrical network. In supply chains, final products are stored at distribution centers before they are selected and packaged to fill customer orders.
- Trucks, supply chains, and distribution systems are still not clearly connected, however. By explaining this relationship in the first few phrases of the introduction, one ought to make it deeper.
Why are you attempting to solve the truck distribution of products issue? Please explain the connection of trucks to other forms of transportation in brief
- Please avoid terms like "dramatic"
- There are two recognized risk categories in supply chains: NN risks and PN risks. Instead of "exceptional events," you should use the phrase "NN risks"; it is difficult to control NN risks because they have unknown probabilities, ranges, and types of consequences
- At the end of the introduction, it states: "...the presented solution was not evaluated with real-world data". Can this help practitioners in any way? I hope it can because science without practical implications is not justified. Please, explain briefly.
- In the conclusion section, one of the claims is that the developed model can be applied to other areas "real-life domains such as healthcare, forest monitoring, and railways". What is meant by forest monitoring? After all, this statement is questionable considering that at the end of the introduction, you say that the research is not related to real-world data.

Experimental design

- You say that the main goal that algorithms should achieve is to allocate all arriving trucks to available warehouses in order to ensure that all trucks are unloaded in the shortest possible time. Have you defined the longest waiting time? Have you taken into account the preferences of the goods to be unloaded, e.g., food, fruit, and perishables?
- What do you exactly mean when you say that "trucks waiting in the warehouse yard compete to unload their cargo in time"?
- How does the used Windows 10 operating system's specified performance affect the obtained results?

Validity of the findings

How practitioners can use the proposed method in real-life problems, and how the proposed method is useful for future studies?
Is the "Proposed Model" a novel/new approach? In its absence, would the results be significantly different?

Additional comments

One of the most beneficial contributions to your research would be to provide some connections with real-life data in context to the main basis for some kind of finished goods supply chain. An explanation regarding how your research can especially assist practitioners would be appreciated. Additionally, it would be useful if you could give a rough timeline for when the suggested improvements can be anticipated.

---

## Round 0.3 · Minor Revisions

Dear authors,

Please make additional minor revisions according to the reviewer's opinion before the official acceptance of the paper.

Reviewer 2 ·

Basic reporting

Dear authors,
I would like to thank you for your valuable answers because they have significantly improved the quality of the research. Although I would suggest that the paper be accepted for publication, I ask that you satisfactorily answer the questions raised in the minor revision. With due respect, please clarify in the final version:

5) Referee’s comments:
There are two recognized risk categories in supply chains: NN risks and PN risks. Instead of "exceptional events," you should use the phrase "NN risks"; it is difficult to control NN risks because they have unknown probabilities, ranges, and types of consequences.

Answer
Due to the nature of the research and the type of problem presented in this work, we decided to avoid delving too deeply into the specialized part of the supply chain issue to keep the reader's attention on the presented problem. Because of this, and in full appreciation of the reviewer's suggestion, we shall continue to use the phrase "exceptional events" with a brief explanation of what is meant by this term.

Comment: Please review and cite the literature in which this term is used in the final version.

Experimental design

3) Referee’s comments:
How does the used Windows 10 operating system's specified performance affect the obtained results?


Answer
It is important to specify the environment of the research work. Windows 10 has an effect on the manager of the process memory.

Comment: Please explain how this affects the quality of the results obtained. How does Windows 10 affect the process memory manager?

Validity of the findings

no comment

Additional comments

Dear authors and editors,
I am pleased to suggest that this work be accepted for publication. Although additional revision is not necessary, I respectfully invite the authors to briefly explain their answers to the two questions raised in the minor revision. These clarifications, in my opinion, will help to further raise the quality of this highly valuable study.

---

## Round 0.4 · accepted · Accept

Dear authors,

Your last revised version of the paper is ready for acceptance.